# Experimental Research on the Process Parameters of a Novel Low-Load Drill Bit Used for 7000 m Bedrock Sampling Base on Manned Submersible

Yu-Gang Ren [1,2], Lei Yang [2], Yan-Jun Liu [1,3,*], Bao-Hua Liu [2], Kai-Ben Yu [2] and Jian-Hua Zhang [1,*]

1 School of Mechanical Engineering, Shandong University, Jinan 250061, China; ytrenyugang@163.com
2 National Deep Sea Center, Qingdao 266237, China; ndsc248@163.com (L.Y.); liukunsh@ndsc.org.cn (B.-H.L.); yukb@ndsc.org.cn (K.-B.Y.)
3 Institute of Marine Science and Technology, Shandong University, Qingdao 266237, China
* Correspondence: LYJ111KY@163.com (Y.-J.L.); jhzhang@sdu.edu.cn (J.-H.Z.)

**Abstract:** Due to the need for accurate exploration of deep-sea scientific research, drilling techniques by combining the operational advantages of the Jiaolong manned submersible is considered one of the most feasible methods for deep-sea bedrock drilling. Based on deep sea bedrock cutting model and discrete element simulation, as well as efficient drilling as the design criterion, the development of a deep sea 7000 m electromechanical coring apparatus was carried out. The outstanding feature of this technology is that the bit load produced by the drill pressure is usually within the range 100–400 N while the recommended load for diamond drilling is 1–3 KN or even more. Therefore, searching for the drilling bits that can drill in extremely hard formations with minimal load and acceptable rates of penetration and rotary speed is the necessary step to prove the feasibility of electromechanical deep-sea drilling technology. A test has been designed and constructed to examine three types of drill bits. The results of experiments show that the new low-load polycrystalline diamond compact (PDC) bit has the highest penetration length of 138 mm/15 min under a 300 N load and 250 rpm rotary speed. Finally, field tests with the Jiaolong submersible were used to conduct deep sea experiments and verify the load model, which provides theoretical and technical data on the use of a low-load core sampling drill developed specifically for a deep sea submersible.

**Keywords:** Jiaolong manned submersible; low-load drill; deep sea drilling; coring apparatus

## 1. Introduction

Drilling into the deep seafloor to retrieve bedrock core samples is crucial for many types of scientific research. Drilling has played an important role in international scientific programs, such as the Deep-Sea Drilling Program, the Ocean Drilling Program and the Integrated Ocean Drilling Program, and has promoted the advancement of geoscience research [1]. The information recorded in deep sea cores can help scientists understand the long-term evolution of the deep sea, as well as changes in the atmosphere, the dynamics of plate tectonics and the distribution of organisms in the subsea floor biosphere. The data obtained from such cores have also become central to understanding the potential for ocean mineral exploitation and the development of seafloor mining equipment. However, drilling and retrieving cores from the deep seafloor is an arduous and time-consuming task. Extreme conditions, such as high pressures, lack of light and low temperatures, present major operational challenges.

To recover deep sea bedrock samples, two types of drilling technology might be considered: (1) seafloor-deployed rig system, which is restricted by the ship platform and ocean swells, thus restricting the coring accuracy [2,3]. (2) lightweight drill systems that can be used by a submersible. These drilling technologies have different concepts, limits, performance and applicable scopes. as shown in Figure 1.

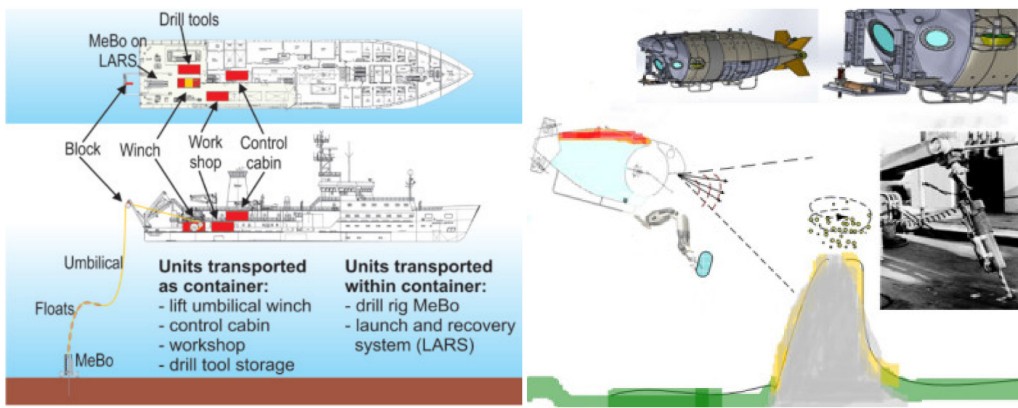

**Figure 1.** Deep sea drilling technology.

To use seafloor-deployed rig in these heavy conditions, many components (e.g., hydraulic system, Launch and Recovery System, etc.) should be largely redesigned as they are not able to realize the core with meter level accuracy in the complex terrain of hydrothermal chimney area and seamounts area [4]. Therefore, developing a powerful, accuracy drilling system has become an important goal for many projects [5,6]. In addition, seafloor-deployed rig systems are still very heavy and power consuming. They require a large logistical load to move and support, so using them in deep sea is not only disadvantageous but also in some cases impractical.

In our opinion, the most accuracy method to penetrate deep sea bedrock is electromechanical cable-suspended drilling technology base on submersible in which the bit is driven directly by a controlled motor to provide power to efficient core [7]. The main advantage of this system is that it combines the fine manipulation capability of a submersible with drilling technology. It is very difficult to conduct insitu coring of hard cobalt crusts. The United States' Alvin manned submersible and the Japanese Shinkai 6500 manned submersible have successfully used a drill to sample high temperature hydrothermal vent chimneys [8,9], followed by the development of Russian, French and British systems for Remote Operated Vehicle (ROVs) that have also successfully cored oceanic cobalt crusts [10,11], Table 1 shows the development status of lightweight drilling systems for use by deep sea vehicles. Zhao [12] proposed a hydraulic core sampling drill design that combined oil cylinder propulsion and pressurization with high-speed hydraulic motor rotation. Gao [13,14] proposed an electric coring drill for the Jiaolong manned submersible driven by a soft shaft. Tian [15] proposed a 4500-m rated drilling system for cobalt-rich crusts investigated the technical difficulties such as bit design and process parameter calculations and conducted sampling tests.

Even though drilling systems have been widely used for conventional bedrock drilling, deep sea core drilling requires that bedrock core be acquired in a limited amount of time, and also that the seabed cores must be accurately collected from a target area. However, the available drilling capacity of a submersible is limited by the deep-sea environment and the carrying capacity of the submersible. As such, the force available for bedrock fragmentation is extremely low, i.e., only one tenth of the drilling pressure available on land [16]. This is called the low-load drill bit operating mode. In this study, a deep-seabed low-load drill system designed with a direct drive integrated mechanics, bit and electronics for use by submersible was developed, constrained by the unique conditions present in the deep-sea environment and the drilling capacity. as shown in Figure 2.

**Table 1.** Development status of lightweight drill systems for use by a submersible.

| Coring Apparatus | Technical Indicators | Country | Core Parameters | |
|---|---|---|---|---|
| ALVIN | 5000 m, 35 kg, Hydraulic drive | USA | Collect a bedrock sample of 480 mm in length | 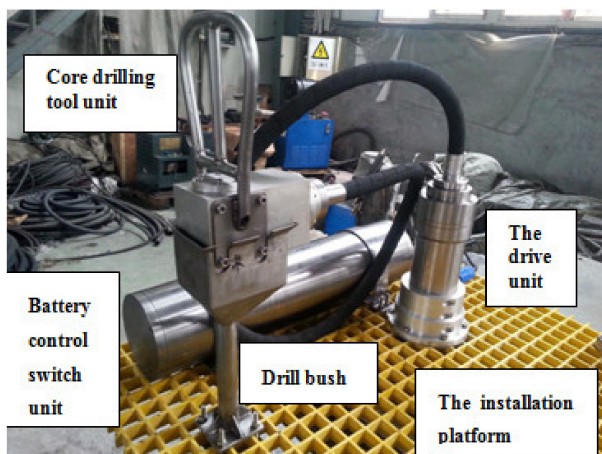 |
| Harbor Branch | 7000 m, Hydraulic drive | USA | Multiple bedrock samples can be drilled at one time | |
| APTYC | 5000 m, 1160 mm, 33 kg, Hydraulic drive | Russia | The diameter and length of the core are 21 mm and 80–130 mm. Cores 5 pieces at a time | |
| CONSUB | 3000 m, Hydraulic drive | UK | Specific parameters unknown | |
| ST-1 | 2000 m, Hydraulic drive | France | Can drill into granite, basalt and other hard bedrocks | |
| Shinkai 6500 | 6500 m, Hydraulic drive | Japan | Can drill into granite, basalt and other hard bedrocks and collect a core of 200 mm in length | |

**Figure 2.** The drill system developed for the Jiaolong submersible.

Therefore, searching for the diamond bits that can drill in extremely hard formations with minimal load and acceptable rates of penetration and torque is a necessary step to prove the feasibility of electromechanical deep sea drilling technology. An orthogonal design method was adopted to analyze the influence of the bit structure parameters, including the helical angle, number of blades and the blade length. This paper describes results from a series of experiments to investigate the low-load diamond drilling of rocks with high hardness and abrasivity. Based on the experimental results, a drill bit for the deep-sea drill system was designed and tested by the Jiaolong submersible.

## 2. Materials and Methods

### 2.1. Bedrock Drillability

The mechanical properties of the bedrocks to be drilled should be known. However, our current knowledge of the deep sea is limited by current technology [17,18]. The deep seabed contains different bedrock types, such as cobalt crusts and polymetallic sulfides, the specific mechanical properties of which are not clear. Cobalt-rich crusts are mainly distributed on the tops and slopes of seamounts at 800–3000 m depths. These crusts are composed of iron–manganese oxides and hydroxides [19], with surfaces that are kidney-shaped, oolitic or nodular, black or dark brown in color and have layered or dendritic cross-sectional structures. The thicknesses of these crusts are generally 5–6 cm, with an

average of ~2 cm, while the maximum thickness can reach ~10–15 cm. The crusts can form on substrates such as basalts, vitreous clastic basalts and montmorillonite [20,21].

Deep sea bedrock drillability can be classified into 12 grades according to the bedrock hardness values. The hardness of cobalt crust bedrock was measured by bedrock pressure hardness tester shown in Figure 3 and the drillability of the samples is listed in Table 2. The indentation hardness index can be calculated as follows:

$$H_y = \frac{P.S}{A} \tag{1}$$

where $H_y$ is the indentation hardness index of the bedrock (kgf/mm$^2$; 1 kgf/mm$^2$ = 9.8 mPa), $P$ is the maximum value measured by the hardness tester when the bedrock breaks (kgf/mm$^2$), $S$ is the area of the jack piston that is pressed into the hardness tester (mm$^2$) and $A$ is the contact area between the indenter and the bedrock sample (mm$^2$).

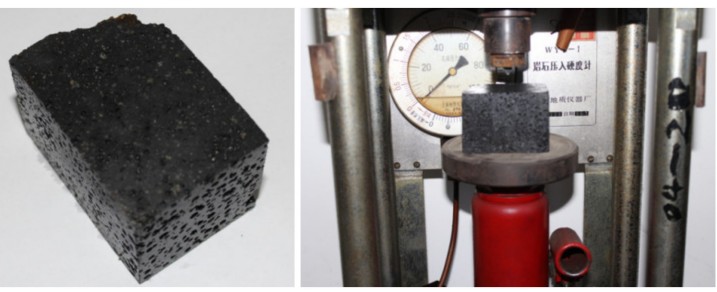

**Figure 3.** Deep sea cobalt crust samples and the bedrock hardness testing apparatus.

**Table 2.** Indentation hardnesses and drillability classification of the bedrock.

| Bedrock Classification | Pressure Hardness Index | | | | Drillability of the Bedrock |
|---|---|---|---|---|---|
| | Test Value (kgf/cm$^2$) | | | Calculated Value (kgf/mm$^2$) | |
| Cobalt crust | 22 | 22 | 23 | 202.5 | 6 |

### 2.2. Bedrock Fragmentation Mechanism and Cutting Force Analyses

Since the diamond bedrock fragmentation mechanism is complex, widely considered drilling process is similar to a grinding wheel, where a microscopic single diamond grain acts as a small blade or tooth. When the drill grinds the bedrock, more sharp edges and abrasive corners come into contact with the surface of the bedrock due to the horizontal drilling force and pressure. Due to extrusion and deformation of the bedrock surface, which is greater than the coupling force between the bedrock particles, the bedrock separates from the bedrock cuttings, producing bedrock fragmentation [22].

According to an analysis of the bedrock crushing mechanism, a single diamond is assumed to be a plastic ball with an inelastic mean value under microscopic conditions (as shown in Figure 4) with radius *r*. Considering that the curvature of the drilling path is much greater than the radius of the diamond itself, the grinding path is assumed to be a straight line and the bedrock mechanical failure process can be determined according to the Coulomb criterion.

(1) Determination of minimum drilling pressure.

Under horizontal and axial loads, the diamond blade cuts into the bedrock to form cutting points at different locations. Location 1 is the external friction state, resulting in elastic deformation of the bedrock and no brittle fracture. At location 2, brittle bedrock fracture, tensile fracture, shear fracture and a shear body are generated. At location 3, diamonds slide along the bedrock layer where the previous diamond was broken. At location 4, diamonds shear the bedrock that has been softened by tensile stress.

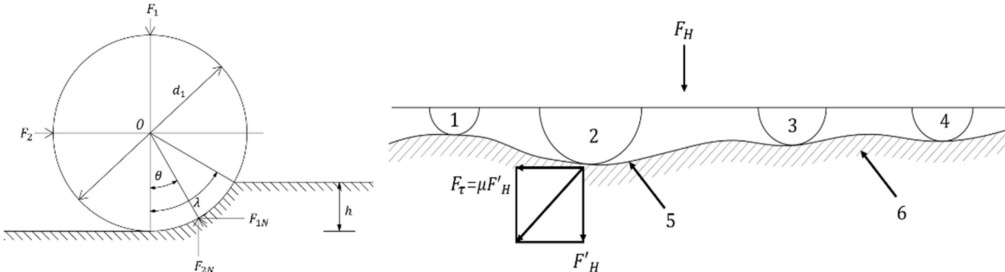

**Figure 4.** Schematic diagram of the bit and interacting with the bedrock interaction.

The four crushing stages can be simplified to a slip field problem of a two-dimensional round punch. Using the research results of R.Hill [23], the normal stress between the diamond bit and the bedrock, $\sigma(\theta, \alpha)$, can be expressed as:

$$\sigma(\theta, \alpha) = \hat{\sigma}\left[1 - \left|\frac{2\alpha}{\pi}\right|\right]\left[A_1 e^{\beta(\pi - 2\theta)} - A_2\right] \tag{2}$$

$$\hat{\sigma}_0 = \sigma_T + \sigma_0\left(\frac{sin\phi}{1 - sin\phi}\right) \tag{3}$$

where $\hat{\sigma}_0$ is the in-situ compressive strength of the bedrock, $\sigma_T$ is the compressive strength of the bedrock, $\sigma_0$ is the pressure around the cutting point, $\phi$ is the angle of friction in the bedrock and $\beta$ is the coefficient of internal friction of the bedrock, where $\beta = tg\phi$.

$A_1$ and $A_2$ are the two relation coefficients and can be obtained from the following:

$$A_1 = \frac{(1 + sin\phi)}{2sin\phi} \tag{4}$$

$$A_2 = \frac{(1 - sin\phi)}{2sin\phi} \tag{5}$$

The single-grain diamond axial pressure can be derived from the formula [24]:

$$F_1 = F_{1N} + F_{1S} \tag{6}$$

$$F_{1N} = \frac{\pi\left(\frac{d_0}{2}\right)^2 \hat{\sigma}_0}{8}\left\{\frac{A_1 e^{\pi\beta}}{\beta^2 + 1}\left[1 - e^{-2\beta\lambda}(\beta sin2\lambda + cos2\lambda)\right] + A_2(cos2\lambda + 1)\right\} \tag{7}$$

$$F_{1s} = \frac{\mu_0\left(\frac{d_0}{2}\right)^2 \hat{\sigma}_0(\pi J_1 - 2J_2)}{2\pi}\left\{\frac{A_1 e^{\pi\beta}}{\beta^2 + 1}\left[\frac{1}{\beta} - e^{-2\beta\lambda}\left(\beta sin^2\lambda + cos2\lambda + \frac{1}{\beta}\right)\right] + A_2(sin2\lambda - 2\lambda)\right\} \tag{8}$$

where $F_1$ is the axial pressure acting on the diamond, $F_{1N}$ is the normal stress component, $F_{1s}$ is the shear stress component, $d_a$ is the diameter of the diamond particle, $\lambda$ is the cutting contact angle, $\mu_0$ is the sliding friction coefficient between the diamond and the bedrock and $J_1$, $J_2$ are coefficients determined by:

$$J_1 = \int_0^{\frac{\pi}{2}} cos\delta_g da \tag{9}$$

$$J_2 = \int_0^{\frac{\pi}{2}} \alpha cos\delta_g da \tag{10}$$

We then calculated the minimum drilling force. After the minimum axial pressure on each diamond is obtained, the minimum pressure required for drilling can be obtained as long as the number of diamond particles involved in cutting on the bit tip is known. The normal stress component obtains the following conclusion through a large number of

experiments; in other words, the number of diamond particles involved in cutting on the bit tip account for ~6–8% of the total number of diamond particles in the layer. We used 7% for our calculations.

Therefore, the minimum drilling pressure required is $P_{min} = F_1 \times F_T$. Based on this, the minimum boundary conditions of drilling pressure under different diameters can be determined, such as the minimum drilling pressure required for a 30-mm bit is 150 N.

(2)    Determination of Rotary Speed Range.

The drilling rate can then be calculated. The normal drilling process can be expressed by the following inequalities:

$$h_1 \leq h_H \leq h_3 \tag{11}$$

where $h_1$ is the micro-cutting depth and $h_3$ is the cutting depth corresponding to severe wear on the diamond, and:

$$\gamma = \frac{2F_T}{A} = \frac{2\mu_c F'_H}{\pi d_a h} \tag{12}$$

where $F_T$ is the shear stress, $A$ is the diamond cutting area, 2 is the load factor and $\mu_c$ is the diamond motion resistance coefficient. If we substitute $\gamma$ into Equation (12) we obtain:

$$h_1 = h_2 = \frac{d_a}{8} + \sqrt{\frac{d_a^2}{64} + \frac{\mu_c F'_H}{\pi \sigma_T}} \tag{13}$$

Suppose Equation (14) is the derivative of some state function $f(h)$ during drilling:

$$f(h) = \int \left( h^2 - \frac{d_a}{4}h + \frac{\mu_c F'_H}{\pi \sigma_T} \right) ah \tag{14}$$

We can integrate this to obtain:

$$f(h) = \frac{h^3}{3} - \frac{d_a}{8}h^2 + \frac{\mu_c F'_H}{\pi \sigma_T} h \tag{15}$$

We can then use the second derivative to study this relationship:

$$f' = h^2 - \frac{d_a}{4}h + \frac{\mu_c F'_H}{\pi \sigma_T} \quad f'' = 2h - \frac{d_a}{4} \tag{16}$$

The analysis of the second derivative shows that, when $h > \frac{d_a}{8}$, then $f''(h) > 0$ and when $h < \frac{d_a}{8}$, then $f''(h) < 0$.

Therefore, the boundary conditions for normal drilling is

$$\frac{d_0}{8} - \sqrt{\frac{d_a^2}{64} - \frac{\mu_c F'_H}{\pi \sigma_T}} < h < \frac{d_a}{8} \tag{17}$$

For the above drilling boundary conditions, according to К.Г. Володченко [24] the thickness of the cut bedrock can be calculated using the following formulas:

$$f = 0.28^5 \sqrt{\frac{2}{d_a} \left[ \frac{a_p \left( D_0^2 - D_1^2 \right) \left( 1 - \left( \frac{d_a}{2} \right) \sqrt[3]{Z} \right) V_M}{(a_p - 1) Z D_C^3 b_t n} \right]^2} \tag{18}$$

$$F'_H = \frac{F_H}{Z_T} \tag{19}$$

By combining Equations (18) and (19), we can obtain the Rotary Speed that will ensure normal drilling:

$$\frac{d_0}{8} - \sqrt{\frac{d_a^2}{64} - \frac{\mu_c F_H'}{\pi \sigma_T}} < 0.28^5 \sqrt{\frac{2}{d_a} \left[ \frac{a_p \left(D_0^2 - D_1^2\right)\left(1 - \left(\frac{d_a}{2}\right)\sqrt[3]{Z}\right) V_M}{(a_p - 1) Z D_C^3 b_t n} \right]^2} < \frac{d_a}{8} \tag{20}$$

### 2.3. Low-Load Drill Bit Design

Based on the special physical and mechanical characteristics of deep-sea rocks, special work requirements have been put forward on the design of the core drill bit: the core remover is carried by the Jiao Long manned submersible, and the submerged manipulator grips the core remover for drilling operations Therefore, the design characteristics and process performance of deep-sea special drills have the following requirements: (1) The drill has high cutting efficiency, powder discharge ability, abrasion resistance. (2) Low power consumption. Under certain drilling rules, drilling consumes less power. (3) The drill bit has the ability to break through the hard rock layer (level 6), and there is no phenomenon of bit passivation and burning. After determining the drillability of the bedrock intended to be drilled, we conducted a mathematical mechanics analysis of the axial force, tangential force and the cutting power of the drill bit and proposed a single-ring four-tooth uniformly distributed diamond composite chip (PDC) drill bit design.

### 2.3.1. Design of Negative Rake Angle

The negative rake angle design can not only improve the working rigidity and cutting speed of the composite sheet, but also extend the service life of the composite sheet, reasonable design can enhance the ability of chip removal, increase the drilling pressure, reduce the residence time of cuttings and the front edge surface of composite sheet, reduce the working temperature of composite sheet, friction coefficient and the resistance of cuttings flowing along it.

The size of the negative rake angle depends on the formation. For the formation with good drillability, bit life and aging, the negative rake angle of the cutting block of the composite bit is 0~5°. For most sedimentary rocks, drill bit performance is best when the negative rake angle is generally −10~−20°. Based on the recommended parameters in the drilling manual and engineering test experience, we calculated the negative rake Angle as 15.

### 2.3.2. Design of Bypass Angle

Since the PDC bit has a small contact area at the bottom of hole, high specific pressure, wear-resistant composite sheet, self-sharpening and sharp edge, it has high efficiency and many cuttings, so it is very important to remove cuttings at the bottom of hole in time. Therefore, a reasonable design should be a certain angle of the composite chip cutting block in the counter-rotation direction when positioning on the drill bit (Figure 5), which is called the bypass angle of the composite chip cutting block. When the bit is rotated for drilling, the cuttings move outwards under the push of force F, which helps to leave the front edge surface and is conducive to timely removal. At the same time, there are also unfavorable factors such as reducing the effective cutting force of the drill bit and increasing the difficulty of the welded composite piece, so the bypass angle should not be designed too large and the bypass angle should be 5° through the test.

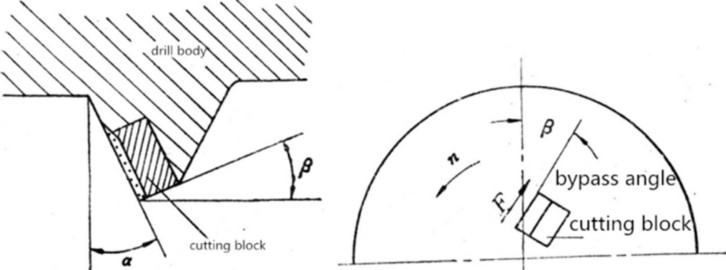

**Figure 5.** The negative antecedent Angle and bypass Angle of the composite chip bit.

### 2.3.3. Exposure Height

From the point of view of formation of cuttings, the space (exposed height) for cuttings should be as large as possible to avoid being hindered and squeezed when the cuttings flow along the front edge surface, as shown in Figure 6, However, when the exposed height is too high, the chip removal speed will be reduced, while when the exposed height is too low, the cuttings flow will be blocked. In order to avoid the problems such as repeated cuttings fragmentation, poor cuttings circulation and overheating of composite plates, the drill bit adopts a high exposure design scheme. The exposure height is optimized to be 3 mm through the test.

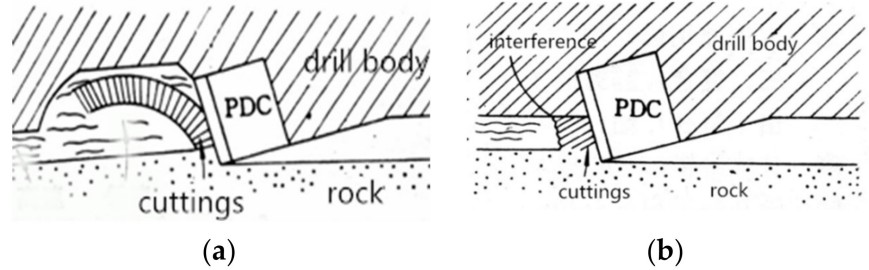

(a)   (b)

**Figure 6.** Exposure height of PDC. (**a**) Removal speed reduced by high exposure; (**b**) Flow obstruction by low exposure.

Based on the above research, a low-load sampling bits are mostly used in space-based applications. Based on the structural design of bits used in space, a novel low-load, high-efficiency bedrock coring bit was designed according to the physical properties of deep-sea bedrock. The newly developed drill adopted a double row distributed discrete cutting edge, both inside and outside, set in a spherical helix spiral and cut off by a ring at the drill pipe junction. This reduces the influence of the cutting edge on the bit matrix spiral chip channel and produces fractured bedrock by cutting, grinding and breaking, also improving the drilling efficiency under the limited drilling capacity and stability. The cutting-edge configuration is shown in Figure 7. The sharp angle of the cutting edge first cuts into the bedrock as a point contact, such that the drilling tool can quickly center itself and maintain drilling stability. The sharp edge consists of two unit cutting edges that are used to cut broken bedrocks during drilling. In order to ensure that the cylindrical shaped core produced after drilling enters the core-collecting mechanism smoothly and without affecting the inner rotation of the core-collecting soft belt, an inner edge was designed at the sharp angle edge of the inner row and the bit barrier ring. The cutting envelope diameter formed by the inner edge of the three angular vertical edges is slightly smaller than the inner diameter of the bit barrier ring, making the outer diameter of the cut core slightly smaller than the inner diameter of the bit core channel, preventing the core from blocking the core channel during drilling.

Based on the theory of discrete element mechanics, we used the EDEM discrete element particle flow simulation software to model and analyze the core sampling behavior. According to the situation, we conducted multi-body coupled joint simulation analyses

based on Fluent + EDEM + SolidWorks, as shown in Figure 8, to achieve more effective and realistic modeling and analyses.

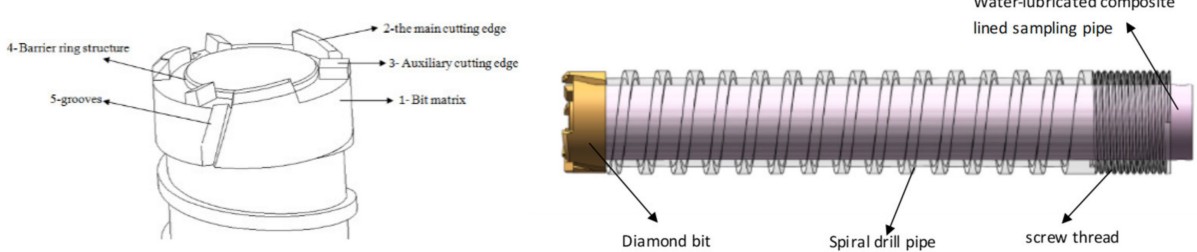

**Figure 7.** The novel low-load drill bit developed for the Jiaolong manned submersible.

**Figure 8.** Analysis of the deep-sea drilling method described in this study using the discrete element method.

The EDEM discrete element simulation method was used to verify the drilling characteristics of the drill bit for chip removal and coring. The number of particles discharged by the bit (chip removal efficiency) was used to evaluate the chip removal capability of the bit. The coring ability of the drill was evaluated from the coring rate at a limited drilling depth. Through a comparative analysis of two types of drill bits used in simulated drilling, the differences in chip discharge and coring characteristics of the bits before and after inlaying the cutting edge were obtained. According to the theoretical analyses and calculations, the internal and external diameters of the drill bit used were Φ31 mm and 35 mm, respectively, and the width of the bottom lip of the drill was 4.5 mm.

Meanwhile, during the drilling of cemented carbide drills, the pressure and power required for drilling increase as the cutting angle increases. When the cutting-edge angle is greater than 50°, the ratio of the required pressure and power increases significantly, indicating that the cutting-edge angle should not be greater than 50°.

### 2.4. Drill Bits

Three types of drill bits (Figure 9) were used to core the test bedrock samples: A1, A2 and A3 standard impregnated diamond drill bits with different matrix harnesses and face contact areas.

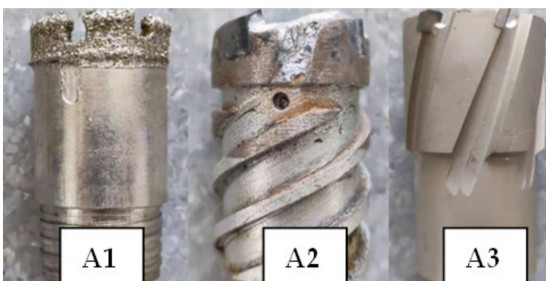

**Figure 9.** Drill bits used on the test bedrock samples.

Two different types of coring bits and the new low-load bit were selected for comparative study and to verify the advantages and disadvantages of the bits in the same conditions. A1 was a traditional impregnated diamond drill bit which widely used on land. A2 was the new low-load Polycrystalline Diamond Compact (PDC) bit and, A3 was a carbide bit used for drilling cores in the field of mechanical processing. Testing this third bit was mainly accomplished as an exploratory test. The basic parameters of tested drill bits are given in Table 3.

**Table 3.** Parameters of tested diamond drill bits.

| Number | Bit Type | OD/ID (mm) | Matrix Hardness (HRC) |
|:---:|:---:|:---:|:---:|
| A1 | Impregnated diamond drill bit | 45/30 | 40–50 |
| A2 | New low-load PDC bit | 45/30 | 40–50 |
| A3 | carbide | 45/30 | 40–50 |

### 3. Results

Base on bedrock fragmentation mechanism force analyses and coring experiment, The recommended parameters for drilling a level 6 cobalt crust to a depth of >100 mm is drilling pressure (0–400 N), drill bit diameter (0–40 mm), rotary speed (0–400 rpm), mechanical properties of the bedrock (this paper is mainly based on cobalt crust) [25].The bit load (often referred to as weight-on-bit or drilling pressure and the rotary speed applied to the drill bit is one of the most important variables in achieving the desired penetration rate and optimizing the life of the bit. For this experimental study, the bit loads range from 90 N to 380 N were used. When the load is insufficient, the diamonds on the bit become polished,

requiring the matrix to be stripped to expose a new layer of diamonds, resulting in very slow penetration.

The rotation of the drill bit causes the exposed cutting element to cut into the bedrock mass. Diamond drill bits require higher rotary speeds than other types of coring drill bits to achieve an acceptable penetration rate. The recommended rotational velocity of the outer diameter of a diamond bit is 140–240 m/min. In general, higher rotational velocities will yield faster penetration rates. On the other hand, a high rotary speed also increases the required torque, the run-out of the core barrel and the hydraulic resistance due to the rotation of the core barrel in a liquid. Therefore, in this study, three rotation speeds of 100 rpm, 200 rpm and 250 rpm were chosen. The drilling experiment setup is shown in Figure 10.

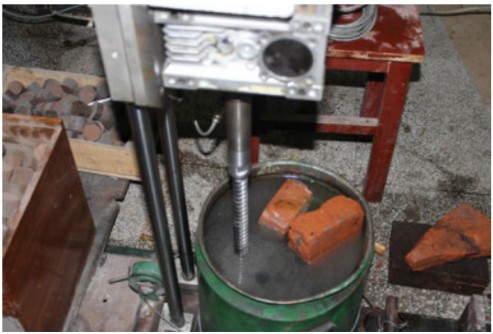 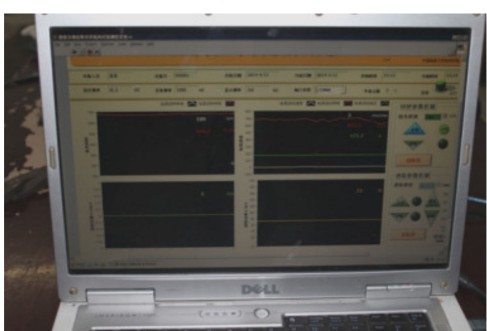

**Figure 10.** The setup of the drilling experiment.

The coring experiment was conducted using the three bits (A1, A2 and A3) at a constant rotation speed of 200 rpm and a working time of 15 min by changing the drilling pressure. The results of the experiment are shown in Figure 10. The coring performance of the A2 bit is better in the simulated deep-sea environment, followed by A1 and A3.

As shown in Figure 11, Compared with A1 and A3 bits, A2 bit has the following advantages. (1) The A2 bit adopts two rows of distributed discrete cutting edge inside and outside, which reduces the influence of cutting edge on the spiral chip removal channel of bit matrix and improves the drilling efficiency and thermal safety under Limited drilling capacity. (2) A1 impregnated diamond bit has flat drilling surface, so it is difficult to achieve rapid centering and maintain drilling stability in deep sea environment. (3) A3 carbide bit is mainly made of tungsten, cobalt, iron and other metal materials. In the process of drilling, the sharp teeth are easily blunt by the rock, which is equivalent to increasing the cutting-edge angle of the cutting teeth, which is not conducive to continuous drilling. (4) The cutting tooth of A2 bit is micro powder polycrystalline diamond composite, which has high strength and high wear resistance. At the same time, the composite bit has self-sharpening ability and sharp blade, so it can achieve high speed drilling with low cutting tools and low speed band and can maintain the effect of high-speed drilling in the whole use process.

Through comparative analyses, we concluded that the A2 bit has advantages over the other two-bit types tested in this study. Future experimental studies will focus on optimizing the parameters of the A2 bit and analyses of drilling pressure, rotation speed and coring ability will be conducted to obtain the optimal technical parameters for the drilling system.

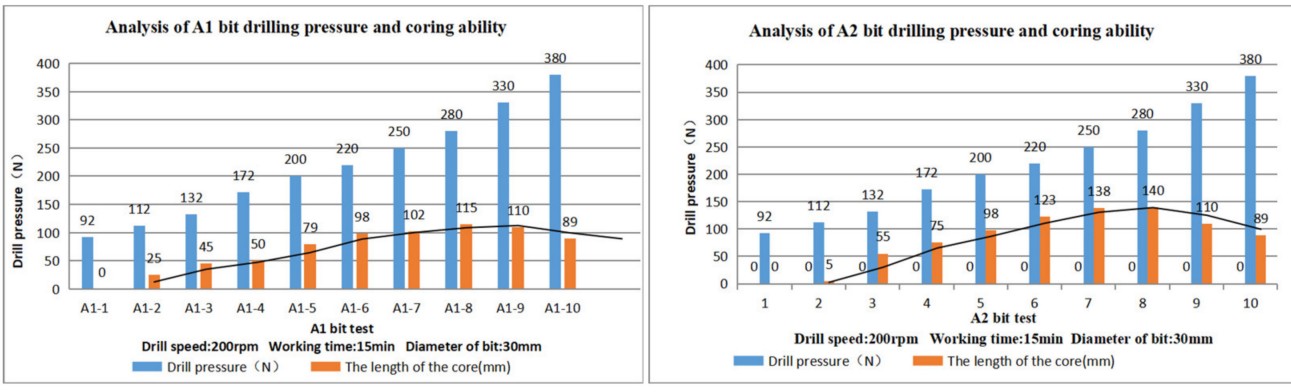

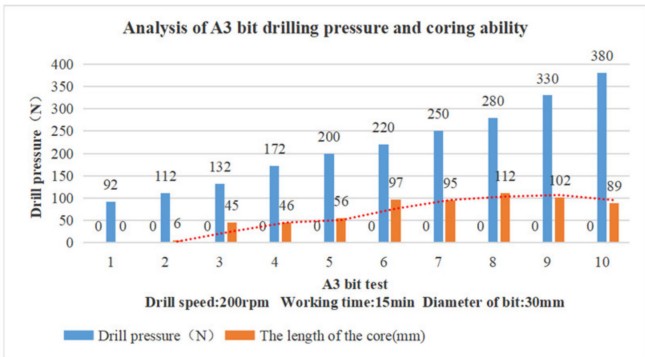

**Figure 11.** Coring tests on the three-bit types at certain speeds.

## 4. Discussion

After determining that the A2 bit has advantages, optimized A2 bit opening test analyses were conducted at variable rotation speeds and drilling pressures. The length of the core obtained from each test was used as the reference for evaluating the coring ability of the bit (Drilling a core of more than 100 mm would be deemed as a successful condition), main time, subject to manned submersible operational capability, the pressure on the drill bit did not exceed 400 N, the rotary speed was held below 400 rpm. Two parameters (drilling pressure and speed) were varied in each double parameter test. Based on the preliminary bit-based tests, the preliminary speed was set as 250 rpm and the drilling pressure was 280 N. These conditions were used as the benchmark test parameters, as shown in Table 4 and Figure 12. The design speed range was 100~380 rpm and the design drilling pressure range was 100~380 N.

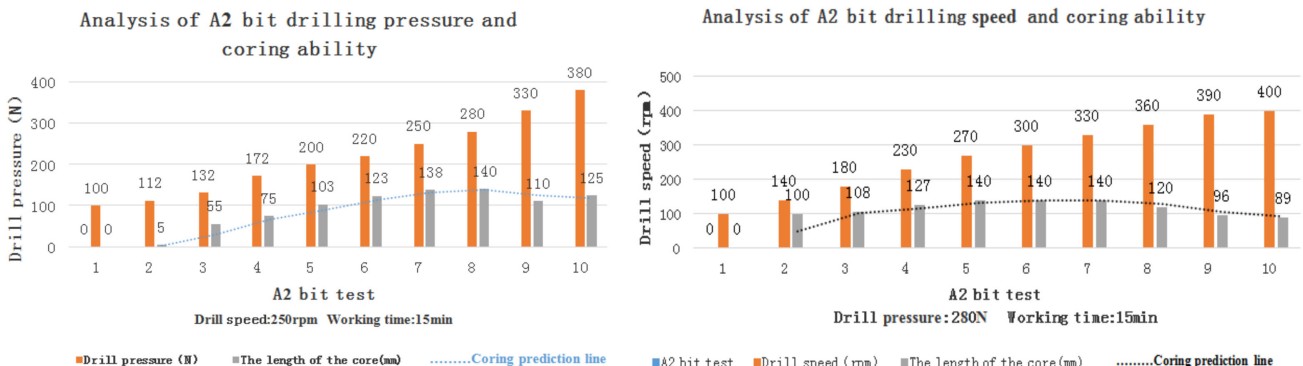

**Figure 12.** Prediction and analysis of drilling pressure, rotary speed and coring ability parameter curve of A2 bit.

According to Table 4, the test results show that the rotary speed at a constant drilling pressure has significant effects on penetration rate, which tends to increase with increases in drilling pressure. The tendency of increase also occurs with the increasing amplitude, but considering that the Jiaolong manned submersible is only capable of providing a maximum of 400 N drilling pressure base on existing data, the drilling pressure used in the experiment could not exceed 400 N.

**Table 4.** Parameters used in the drilling experiment of A2 bit.

| Test Number | Bedrock Category | Parameters and Results of Drilling Test Procedures | | | Torque (N.CM) | Power Test (W) | | Coring Ability | |
|---|---|---|---|---|---|---|---|---|---|
| | | Drilling Pressure (N) | Rotary Speed (rpm) | Drilling Time (min) | Drilling | No-Load | Drilling | Core Length (mm) | Coring Rate (%) Max 140 mm |
| 1 | Cobalt crust | 112 | 250 | 15 | 408 | 33 | 43 | 5 | 3.5 |
| 2 | Cobalt crust | 200 | 250 | 15 | 620 | 33 | 130 | 103 | 70 |
| 3 | Cobalt crust | 250 | 250 | 15 | 605 | 33 | 160 | 138 | 98.5 |
| 4 | Cobalt crust | 280 | 250 | 15 | 610 | 33 | 164 | 140 | 100 |
| 5 | Cobalt crust | 380 | 250 | 15 | 634 | 33 | 176 | 125 | 89.3 |
| 6 | Cobalt crust | 280 | 140 | 15 | 404 | 36 | 75 | 100 | 71.4 |
| 7 | Cobalt crust | 280 | 230 | 15 | 613 | 95 | 130 | 127 | 90.7 |
| 8 | Cobalt crust | 280 | 270 | 15 | 621 | 101 | 170 | 140 | 100 |
| 9 | Cobalt crust | 280 | 330 | 15 | 632 | 112 | 175 | 140 | 100 |
| 10 | Cobalt crust | 280 | 390 | 15 | 618 | 123 | 186 | 95 | 67.9 |

The optimized parameter range of A2 bit (rotary speed 200~350 rpm, drilling pressure 200~300 N) can be obtained basically. From the rotary speed and drilling pressure parameter tests, selected some typical coring parameters, listed in Table 5 and Figure 13. These data show that the minimum power was 75 W (at 140 rpm, 280 N) and the maximum power was 175 W (at 330 rpm, 280 N), for which the range of core lengths was 115~140 mm. Thus, the reasonable parameter intervals are a speed of 270~330 rpm and a drilling pressure of 172~280 N.

**Table 5.** Double parameter tests of Rotary Speed and drilling pressure (A2 bit).

| Test Number | Bedrock Category | Parameters and Results of Drilling Test Procedures | | | Torque (N.CM) | Power Test (W) | | Coring Ability | |
|---|---|---|---|---|---|---|---|---|---|
| | | Drilling Pressure (N) | Rotary Speed (rpm) | Drilling Time (min) | Drilling | No-Load | Drilling | Core Length (mm) | Coring Rate(%) Max 140 mm |
| 1 | Cobalt crust | 92 | 100 | 15 | 256 | 33 | 36 | 0 | 0 |
| 2 | Cobalt crust | 112 | 140 | 15 | 324 | 36 | 40 | 5 | 3.5 |
| 3 | Cobalt crust | 132 | 180 | 15 | 389 | 87 | 76 | 55 | 39.3 |
| 4 | Cobalt crust | 172 | 230 | 15 | 578 | 90 | 121 | 98 | 70 |
| 5 | Cobalt crust | 200 | 270 | 15 | 589 | 101 | 145 | 115 | 82.1 |
| 6 | Cobalt crust | 220 | 300 | 15 | 613 | 107 | 159 | 138 | 98.6 |
| 7 | Cobalt crust | 250 | 330 | 15 | 624 | 112 | 167 | 140 | 100 |
| 8 | Cobalt crust | 280 | 360 | 15 | 638 | 118 | 169 | 110 | 78.6 |
| 9 | Cobalt crust | 330 | 390 | 15 | 641 | 123 | 178 | 99 | 70.7 |
| 10 | Cobalt crust | 380 | 400 | 15 | 617 | 116 | 186 | 78 | 55.7 |

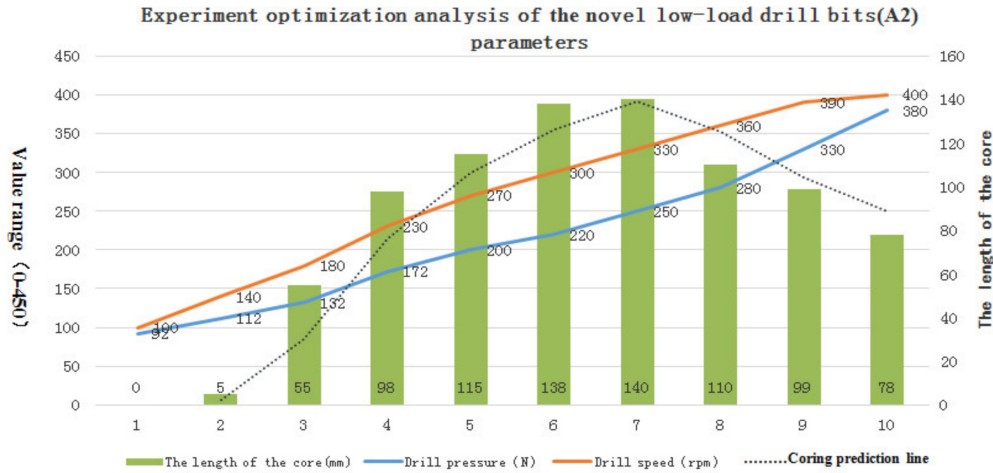

**Figure 13.** Double parameter tests of Rotary Speed and drilling pressure (A2 bit).

## 5. Conclusions

From June to July of 2016, China's 7000 m Jiaolong manned submersible conducted Leg II of the 37th China Dayang Cruise (Figure 14) in the southwestern region of the Mariana Trench, approximately 200 km southwest of Guam (11°22′ N, 142°25′ E). After the new drill system underwent laboratory verification, it was loaded onto the Jiaolong submersible to conduct coring operations. During the two-month expedition, the Jiaolong submersible was deployed for 9 dives, completing numerous tasks that included discovering a mud volcano on the southern slope of the Mariana Trench, conducting investigations of the geology, biology and geochemistry of the region, and bedrock sampling.

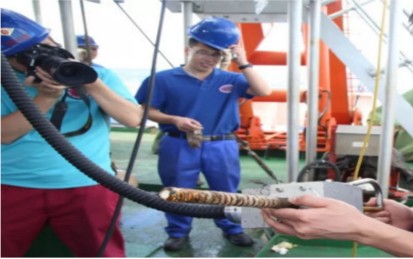
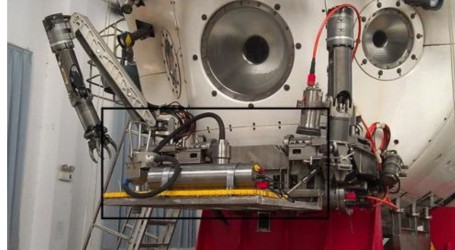

**Figure 14.** The drilling apparatus on the Jiaolong submersible.

The drilling system developed for use by Jiaolong manned submersible includes the low-load coring drill, a battery and control system cabin, a start-stop trigger magnet, a soft shaft drive unit, a clamping handle, a drill sleeve and a mounting tray.

The Jiaolong submersible conducted its 102nd and 104th dives using the coring device developed in this study (Figure 15). The system required approximately 20 min to drill a sample, and core samples of ~14 cm in length were obtained from the seafloor.

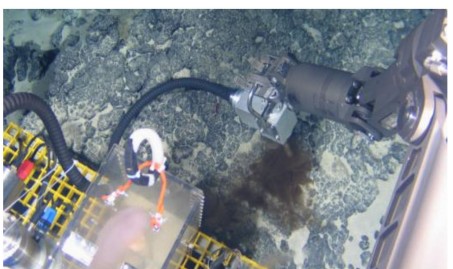
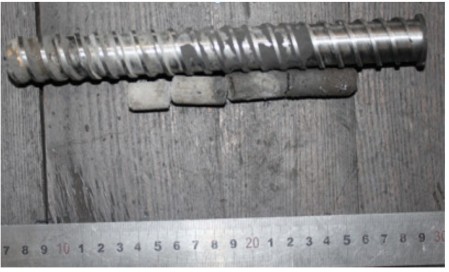

**Figure 15.** Deep-sea drilling operation and core collected from a cobalt crust on the seafloor.

The experimental results show that the drilling pressure has a large influence on the Rotary Speed in the same bedrock sample. When the drilling pressure is constant, the Core length is approximately linearly proportional to the rotary speed.

Due to the hosting capabilities limitations of the Jiaolong submersible, the rotary speed could not exceed 400 rpm and the drilling pressure could not exceed 400 N. The experimental analyses showed that the minimum power was 75 W (140 rpm, 280 N) and the maximum power was 175 W (330 rpm, 280 N) such that the system would meet the power requirement. The core lengths obtained ranged from 100–140 mm for speeds of 270~330 rpm, drilling pressures of 172~280 N, such that the system would meet the power requirement.

The laboratory drilling experiments verified that, when the rotary speed is 250 rpm and the drilling pressure is 300 N, a bedrock sample with a length of 138 mm can be obtained within 15 min of drilling time, which meets the requirements for underwater coring.

The scientific drilling experiments of Jiaolong manned submersible show that this novel low-load bit meets the design requirements, it can provide a strong guarantee for the seafloor crusted rock fixed point core. Future can perform bottom-sitting and hovering operations on the complex seafloor base on Jiaolong submersible, thus ensuring that the corer can reach almost all cobalt crusted mining areas to complete core operations, providing comprehensive and reliable sample data for the development and exploration of oceanographic minerals.

**Author Contributions:** Conceptualization, Y.-J.L. and J.-H.Z.; methodology, B.-H.L.; formal analysis, Y.-G.R. and L.Y.; resources, K.-B.Y.; data curation, Y.-G.R. and L.Y.; writing—original draft preparation, Y.-G.R. and L.Y.; writing—review and editing, all authors. All authors have read and agreed to the published version of the manuscript.

**Funding:** This work was supported by the project Natural Science Foundation of Shandong Province (No. ZR201910220189), the National Key Research and Development Plan (Nos. 2016YFC0300704, 2017YFC030660) the Shandong Provincial Major Innovation Project (No. 2019JZZY010802).

**Institutional Review Board Statement:** Not applicable.

**Informed Consent Statement:** Not applicable.

**Data Availability Statement:** The data presented in this study are available on request from the corresponding author.

**Acknowledgments:** This work was supported by the National Natural Science Foundation of China (No.61603108), the Taishan Scholar Project Funding (No. tspd20161007), We give special thanks to the crew on the China Dayang Cruise. Valuable sediment data and samples were obtained during this voyage and the data provided were crucial to this work.

**Conflicts of Interest:** The authors declare no conflict of interest.

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
