# Peer review of "Experimental Research on the Process Parameters of a Novel Low-Load Drill Bit Used for 7000 m Bedrock Sampling Base on Manned Submersible"

_jmse, doi:10.3390/jmse9060682_

Round 1
Reviewer 1 Report
The paper documents a great scientific exercise. A theory is presented, a test stand is built, labory tests and field tests are run. All results are documented. However, the results are not really discussed. What do we learn? How can we explain what we found? What does it mean for further field runs? The paper leaves a lot of questions open!
Did the calculations correspond to the findings in the laboratory? Did the findings in the laboratory correspond to the findings in the field test? If not: why?
Was the sample drilled in the laboratory the same which was drilled in the sea?
Why did Bit 2 drill better than bit 1 and 3? What was so special about it?
Author Response
Dear reviewer,
Thank you very much for the careful and professional review of the paper. According to your review opinions, the following attached documents are fed back.
Please see the attachment.

Reviewer 2 Report
The paper is interesting because it concerns the study of new low-load drill bit. However, minor corrections are required.
- Page 2, Table 1
Coring apparatus “CONSUB”. Why the specific parameters are not known?
- Page 3, line 105
“The hardness of cobalt crust bedrock was measured…”
Why this type of bedrock has been analyzed? Did the Authors not think to analyze a different type of bedrock?
- Page 6, lines 217-218
“…drill bit design with a negative rake angle of 15°, a bypass angle of 5°, and an exposure height of 3 mm.”
Please add more information about the geometry of the drill. Why values of negative rake and bypass angle were applied.
- Page 8, line 252
“…Diamond Compact(PDC) bit and . A3 was a carbide…”
Before A3 there should not be “.”
- Page 8, line 260
“The recommended uniaxial load for drilling a level 6 cobalt crust to a depth of >100 mm”
What do such recommendations result from?
- Page 9, figure 9
In the charts, there aren’t coring prediction lines.
- Page 9 and 10
There aren’t titles on the charts.
- Page 11, line 343
“…fixed point core. further can perform..”
It should be “...core. Future…”
- Page 11
After “Reference” there is information related to the formation of references. This information shouldn't be there.
- General attention:
Did the Authors not consider in their research the impact of changing drilling parameters on the generation of vibrations? As the drilling speed increases, the vibration level increases, which can adversely affect the service life of the drill bit.
Author Response

(The authors gave the same response as above.)

Round 2
Reviewer 1 Report
The article has been improved, but the English can still be improved. Some sentences are not complete and sometimes capital letters are used in the middle of the sentence.